# Robust Multimodal Sentiment Analysis of Image-Text Pairs by Distribution-Based Feature Recovery and Fusion

## ABSTRACT

As posts on social media increase rapidly, analyzing the sentiments embedded in image-text pairs has become a popular research topic in recent years. Although existing works achieve impressive accomplishments in simultaneously harnessing image and text information, they lack the considerations of possible low-quality and missing modalities. In real-world applications, these issues might frequently occur, leading to urgent needs for models capable of predicting sentiment robustly. Therefore, we propose a Distribution-based feature Recovery and Fusion (DRF) method for robust multimodal sentiment analysis of image-text pairs. Specifically, we maintain a feature queue for each modality to approximate their feature distributions, through which we can simultaneously handle low-quality and missing modalities in a unified framework. For low-quality modalities, we reduce their contributions to the fusion by quantitatively estimating modality qualities based on the distributions. For missing modalities, we build inter-modal mapping relationships supervised by samples and distributions, thereby recovering the missing modalities from available ones. In experiments, two disruption strategies that corrupt and discard some modalities in samples are adopted to mimic the low-quality and missing modalities in various real-world scenarios. Through comprehensive experiments on three publicly available image-text datasets, we demonstrate the universal improvements of DRF compared to SOTA methods under both two strategies, validating its effectiveness in robust multimodal sentiment analysis.

## CCS CONCEPTS

• **Information systems** → **Sentiment analysis**; *Multimedia information systems*; • **Computing methodologies** → **Artificial intelligence**.

## KEYWORDS

robust multimodal sentiment analysis, low-quality and missing modality, feature distribution, modality recovery, modality fusion

**ACM Reference Format:**
Anonymous Authors. 2024. Robust Multimodal Sentiment Analysis of Image-Text Pairs by Distribution-Based Feature Recovery and Fusion. In *Proceedings of the 32nd ACM International Conference on Multimedia (MM'24), October 28-November 1, 2024, Melbourne, Australia.* ACM, New York, NY, USA, 10 pages. https://doi.org/10.1145/nnnnnnn.nnnnnnn

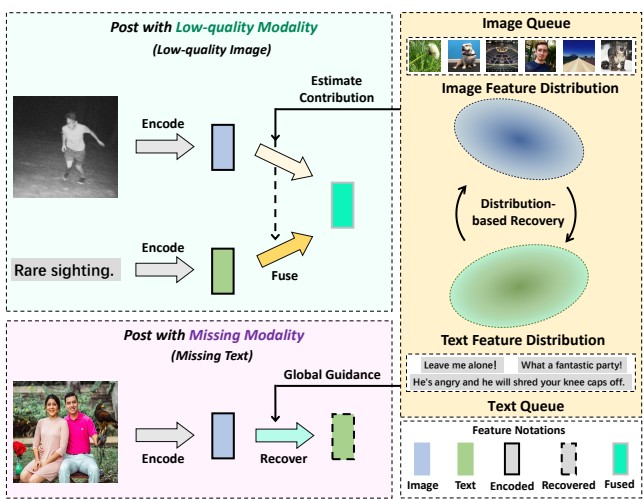

**Figure 1: Brief illustration of DRF. We maintain two feature queues to approximate the feature distributions of images and texts. The distributions can estimate the contribution of each modality for fusion and provide global guidance for modality recovery, facilitating the robustness of the model to both low-quality and missing modalities.**

## 1 INTRODUCTION

With the rapid growth of smartphones, people are getting used to sharing their experiences by posting on social media. In most cases, posts contain information from various modalities. As a result, multimodal sentiment analysis (MSA) that aims to understand the sentiments expressed by users in multimodal content has become a popular research topic. Due to its wide applications in social media analysis [3], recommendation system [24], human-computer interaction [56], and more [1, 55], it attracts substantial attention from both academic and industrial communities [49, 52].

Image-text pairs are a typical form of posts, and analyzing their overall sentiments is an important subfield in MSA. In existing works, the majority seeks to fuse multimodal information by elaborate fusion strategies, such as concatenations [38] and attentional mechanisms [17, 37, 39, 41]. The others attempt to address task-specific challenges, like the ignorance of global co-occurring characteristics [43], modality heterogeneity [35], and data dependency [42, 48]. They achieve impressive progress in fully exploiting information from both visual and textual modalities to model the overall sentiments. However, in real-world applications, the images and texts of posts may be corrupted or missing, leading to frequent occurrences of low-quality and missing modalities. For instance, images are probably pixelated or unavailable due to Not-Safe-For-Work issues and privacy concerns [34], and texts perhaps suffer from information loss or are unrecognizable due to rare languages

and unaligned encoding formats between platforms. These scenarios result in severe performance degenerations of current works, underscoring the necessity of robust MSA methods.

Handling low-quality or missing modalities has been well-studied in related multimodal fields. In trusted multi-view classification [10, 11], researchers assign different weights for each view by estimating its uncertainty to produce reliable predictions with potential low-quality views. In incomplete multimodal learning [18, 27, 44], researchers recover unavailable modalities from the observed ones [34] to enable consistent encoding of samples with arbitrary missing modalities [51]. Despite their success, applying them to handle both issues of low-quality and missing modalities in MSA of image-text pairs would encounter two main challenges. Firstly, the two issues are tackled separately, with unaligned models designed based on distinct strategies, which introduces extra difficulties and alignment burdens for direct combination. Secondly, the user-generated nature of posts from social media results in frequent mismatches between images and texts [43]. This characteristic conflicts with the common assumption in studies on videos or medical images [18, 27, 34], that the information of modalities from the same sample is closely related, impeding the application of these methods.

To fill these gaps, we propose a method called Distribution-based feature Recovery and Fusion (**DRF**), as shown in Fig. 1. We maintain feature queues for images and texts to approximate their respective feature distributions, which enable the model to handle low-quality and missing modalities in a unified framework.

**(1).** For samples with missing modalities, we recover the missing modalities from the available ones by supervising the recovery process based on samples and distributions, thereby encoding them the same as complete samples. The sample-based recovery forces the model to convert between image and text features of the same samples. It effectively builds local connections between modalities, yet is prone to be misled by the mismatches of image-text pairs. Therefore, we introduce an additional distribution-based recovery, facilitating conversion between image and text distributions. Concretely, it encourages the model to predict the mean and variance of one distribution from another. This provides global mapping relationships between modalities and eliminates the negative impacts of the mismatches.

**(2).** For samples with diverse-quality modalities, we determine the contribution of each modality to the fusion based on its correlation with the distribution. Leveraging the global mapping relationships learned by the modality recovery process, we use the recovered modalities that conform to the distributions to compensate for potential low-quality modalities and expand each sample into three. Then, we quantitatively estimate the quality of each modality with Gaussian distribution probability and assign weights for three samples by multiplying the probabilities of its two source modalities. Finally, we compute the overall fused feature as the weighted sum of the three fused features. Through this process, we can dynamically fuse modalities according to their qualities, reducing the influences of low-quality modalities on the fusion.

To systematically assess the robustness of models, we adopt two disruption strategies that randomly corrupt and discard modalities from samples to mimic real-world scenarios of various degrees of low-quality and missing modalities. By conducting extensive experiments on MVSA-S, MVSA-M [25], and TumEmo [41], we prove

the effectiveness of DRF in robust MSA. The main contribution of this paper is summarized as follows:

- We focus on robust MSA of image-text pairs for the low-quality and missing modalities, which are prevalent concerns in real-world scenarios. As far as we know, this is the first attempt to explore the robustness of models in this subfield.
- We propose a novel method, DRF, to handle the low-quality and missing modalities in a unified framework. It leverages two feature distributions to provide global mapping relationships between modalities for feature recovery as well as qualitative estimations of modality quality for feature fusion.
- Experimental results under two disruption strategies on three MSA benchmark datasets demonstrate the significant improvements of DRF compared to the state-of-the-art MSA methods, validating its superiority in robust MSA of image-text pairs.

## 2 RELATED WORKS

### 2.1 Multimodal Sentiment Analysis

Early works on sentiment analysis focus solely on a single modality, such as text [26, 29], image [45, 46] and speech [16, 23]. With the rapid increase of posts in social media, MSA for image-text pairs has garnered increasing attention in recent years. In the beginning, researchers leverage the semantics of images and texts with simple concatenation [37] or attention [38]. Later on, more elaborate attention-based structures are designed to enable more comprehensive modality fusion. COMN [39] iteratively models the interaction between image and text features at multiple levels. MVAN [41] fully exploits the correlations of different views of images and texts. CLMLF [17] leverages Transformer-Encoder [32] for token-level alignments. More recently, the focus of researchers has shifted toward addressing task-specific challenges. MGNNS [43] utilizes graph neural networks to capture the global characteristics of the dataset. MVCN [35] tackles the modality heterogeneity with sparse attention, feature restraint, and loss calibration. UP-MPF [48] and MultiPoint [42] devote to few-shot MSA to avoid annotation costs. There is also a series of studies [14, 19, 40, 47] on fine-grained MSA, aiming to detect the sentiment of a specific aspect within the image-text pair, which though is not the primary focus of this paper.

These methods effectively model the sentiments by relying on complementary information from both images and texts, yet can not properly handle the issues of low-quality and missing modalities. Since these issues might frequently occur in real-life applications [51], we propose DRF, a practical method capable of predicting sentiment for image-text pairs robustly.

### 2.2 Robust Multimodal Learning

The issues of low-quality and missing modalities are prevalent in all types of multimodal data, and various studies have been conducted on them. For low-quality modalities, a feasible strategy is to reduce their influences on the fusion as adopted in trusted multi-view classification [10, 11]. Researchers estimate the uncertainty of each view based on Dempster-Shafer Evidence Theory [5, 28] and give less consideration to the high uncertainty views, which correspond to the low-quality modalities in our case, during the fusion. The uncertainty is also estimated according to other methods

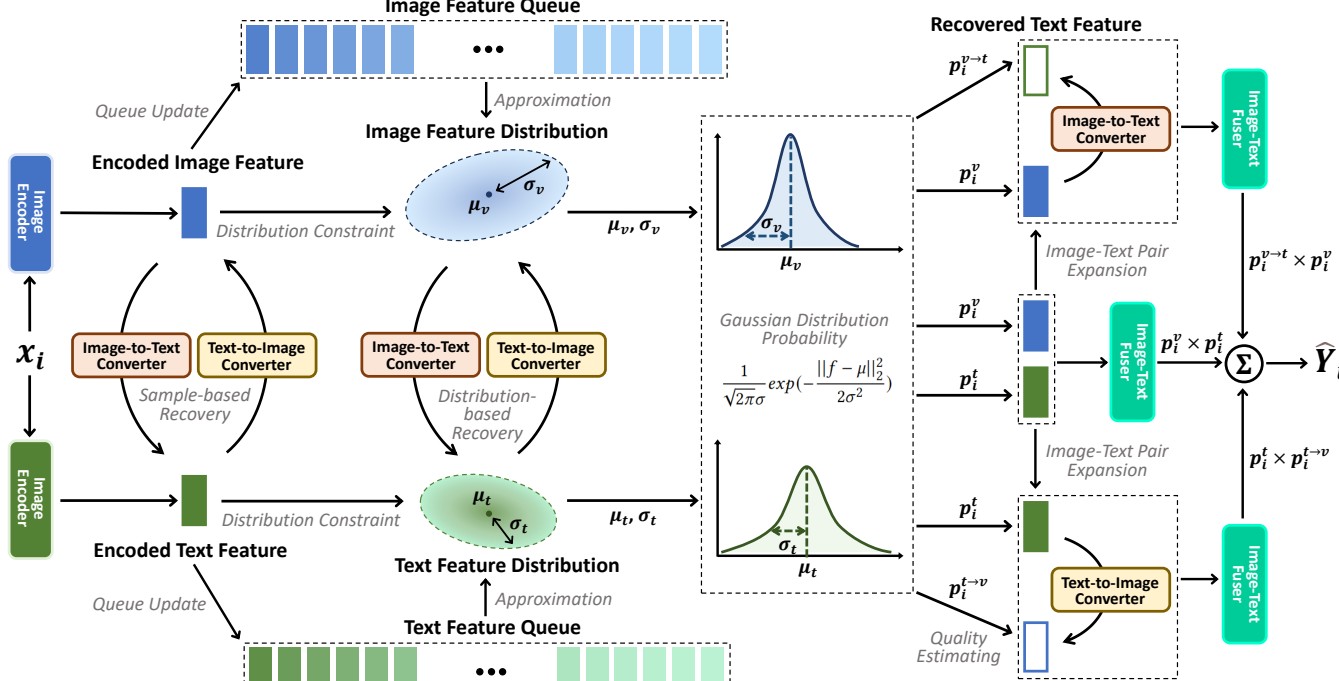

**Figure 2: Illustration of DRF.** The core of our method is the modeling of image and text feature distributions, which we approximate using the respective feature queues. After separate encoding of each modality, we first supervise two converters to learn inter-modal mapping relationships by sample-based and distribution-based recovery. Subsequently, we leverage the recovered features to expand each sample into three. Utilizing the Gaussian distribution probability, we estimate the modality qualities to decide their contributions to the fusion. Finally, we obtain the overall fused feature as the weighted sum of the features of three expanded samples and enqueue features to the queue according to their qualities.

or theories in related studies, including Bayesian neural networks [6, 9], ensemble-based methods [12, 15], Normal Inverse-Gamma distribution [21] and energy score [20, 53]. For missing modalities, data imputation methods [18] in incomplete multimodal learning recover them from the available ones. To achieve this, some researchers directly pad missing modalities with fixed values [4, 50], some others optimize through low-rank projection [2, 22], the rest leverage the generative capability of specific neural networks architectures, such as autoencoder [33] and Transformer [32].

To unifiedly handle both issues in MSA of image-text pairs, we leverage the image and text feature distributions. On the one hand, the distributions can provide quantitative estimations of modality qualities through the probability density function. On the other hand, they can also guide the learning of global mapping relationships between modalities, eliminating the negative impacts of image-text pair mismatches.

## 3 METHOD

### 3.1 Task Formulation

We focus on the sentiment classification of image-text pairs with possible low-quality and missing modalities. We first give a definition of the regular MSA. Given a set of samples $\{(x_i, y_i) | i \in \{1, 2, \cdots, N\}\}$, where $x_i$ denotes the image-text pair $(v_i, t_i)$, $y_i$ is its sentiment label from a total of $S$ categories, and $N$ is the total

number of samples, the model needs to build a mapping between image-text pairs $x$ and sentiment labels $y$.

To simulate the occurrences of low-quality and missing modalities in real-world applications, we randomly corrupt and discard modalities from samples. We denote the discarding operation of image-text pair $(v_i, t_i)$ as $\lambda_i^v, \lambda_i^t \in \{0, 1\}$. Take image $v_i$ as an example: $\lambda_i^v = 0$ represents that it is discarded, in other words, missing, and $\lambda_i^v = 1$ represents the other way. For the corruption operation aimed at simulating low-quality modalities, we consider it invisible to the model because it is also difficult to accurately pre-determine modality quality in practice. Thus, the overall definition of $x_i$ in robust MSA is $(v_i, t_i, \lambda_i^v, \lambda_i^t)$.

### 3.2 Feature Distribution Modeling

The pipeline of DRF is shown in Fig. 2. For convenience, we pretend both the image and text are not discarded while presenting our method and reflect the influences of $\lambda_i^v, \lambda_i^t$ by the computations. After receiving the image-text pair $x_i = (v_i, t_i, \lambda_i^v, \lambda_i^t)$ of an input sample $(x_i, y_i)$, we first encode $v_i$ into image feature $f_i^v \in \mathbb{R}^{d_v}$, and $t_i$ into text feature $f_i^t \in \mathbb{R}^{d_t}$. $d_v, d_t$ are the feature dimensions of the image and text.

In our framework, the core of unified modeling of low-quality and missing modalities is the feature distribution of each modality. To acquire these distributions, limited features from a single

mini-batch are insufficient. Inspired by self-supervised learning [13, 36], we maintain a feature queue for each modality to record features across multiple mini-batches. The feature queue of image is denoted by $Q_v = \{f_j^v | j \in q_v\}$ and it of text is denoted by $Q_t = \{f_j^t | j \in q_t\}$, with the queue size set to $L$ for both of them. By adopting a sufficiently large queue size, we can approximate the feature distributions of all samples by those from feature queues. Specifically, we approximate the mean $\mu_v$ and standard deviation $\sigma_v$ of the image feature distribution by:

$$\mu_v = \frac{1}{L} \sum_{j \in q_v} f_j^v, \tag{1}$$

$$\sigma_v = \sqrt{\frac{1}{L} \sum_{j \in q_v} ||f_j^v - \mu_v||_2^2}. \tag{2}$$

The mean $\mu_t$ and standard deviation $\sigma_t$ of the text feature distribution are approximated similarly.

To encourage the compactness of each distribution and the separation between distributions, we devise a distribution constraint that brings image and text features closer to the means of their respective feature distributions and away from the means of the other:

$$\begin{aligned}\mathcal{L}_{dis} = &\lambda_i^v \cdot exp(||f_i^v - \mu_v||_2 - ||f_i^v - \mu_t||_2) \\ &+ \lambda_i^t \cdot exp(||f_i^t - \mu_t||_2 - ||f_i^t - \mu_v||_2)\end{aligned} \tag{3}$$

### 3.3 Modality Recovery

To handle missing modalities, we build mapping relationships between image and text through two modality converters, which are essentially two-layer MLPs. For the image-to-text converter, denoted by $C_{v \to t}(\cdot)$, an intuitive idea is encouraging it to recover the text feature $f_i^t$ from the image feature $f_i^v$. We call this task sample-based recovery and its loss is given by:

$$\mathcal{L}_{v \to t}^s = \lambda_i^v \lambda_i^t \cdot ||C_{v \to t}(f_i^v) - f_i^t||_2. \tag{4}$$

Its effectiveness is built upon the alignment between information of image $v_i$ and text $t_i$. However, due to the mismatches between images and texts from social media posts [43], such alignment can not be guaranteed for all samples, leading to occasionally negative impacts on the converter. To alleviate these, we devise a distribution-based recovery task that provides mapping guidance from a global perspective. Specifically, we supervise the converter to recover the mean $\mu_t$ and standard deviation $\sigma_t$ of $Q_t$ from $Q_v$. The mean $\mu_{v \to t}$ and standard deviation $\sigma_{v \to t}$ of the converted distribution are computed as:

$$\mu_{v \to t} = \frac{1}{L} \sum_{j \in q_v} C_{v \to t}(f_j^v), \tag{5}$$

$$\sigma_{v \to t} = \sqrt{\frac{1}{L} \sum_{j \in q_v} ||C_{v \to t}(f_j^v) - \mu_{v \to t}||_2^2}. \tag{6}$$

Then, the loss of distribution-based recovery is given by:

$$\mathcal{L}_{v \to t}^d = ||\mu_{v \to t} - \mu_t||_2 + |\sigma_{v \to t} - \sigma_t|. \tag{7}$$

The sample-based and distribution-based recovery tasks are also applied to the text-to-image converter $C_{t \to v}(\cdot)$ with symmetric computations. Thereby, the combined loss of both converters is:

$$\mathcal{L}_{rec} = \mathcal{L}_{v \to t}^s + \mathcal{L}_{v \to t}^d + \mathcal{L}_{t \to v}^s + \mathcal{L}_{t \to v}^d. \tag{8}$$

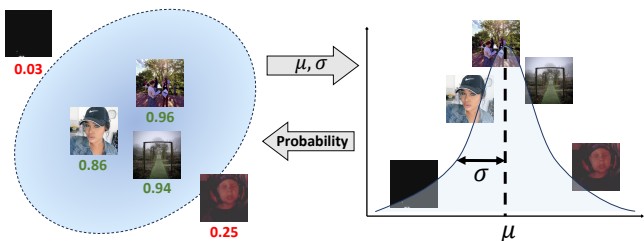

**Figure 3: Examples of estimating image quality based on the feature distribution.**

### 3.4 Modality Quality Estimation

To handle samples with potentially low-quality modalities, we perform multimodal fusion based on the quality of each modality estimated by the feature distributions. Firstly, we expand the image-text pair into three, by treating its image $v_i$ and text $t_i$ as independent samples with missing modalities. Through the modality recovery process, we obtain the recovered image feature $C_{t \to v}(f_i^t)$, denoted by $f_i^{t \to v}$ and the recovered text feature $C_{v \to t}(f_i^v)$, denoted by $f_i^{v \to t}$. Thus, the image and text features of the original sample are $(f_i^v, f_i^t)$, those of the image are $(f_i^v, f_i^{v \to t})$, and those of the text are $(f_i^{t \to v}, f_i^t)$.

Subsequently, we estimate the quality of each modality according to its correlation with the respective feature distribution. We consider those unimodal features that conform to the feature distribution to come from high-quality modalities, while the others to come from low-quality modalities. We adopt the Gaussian distribution to provide quantitative estimations. Its probability density function given feature $f$, mean $\mu$ and standard deviation $\sigma$ is:

$$p(f, \mu, \sigma) = \frac{1}{\sqrt{2\pi}\sigma} exp(-\frac{||f - \mu||_2^2}{2\sigma^2}). \tag{9}$$

We compute the contributions of $f_i^v$ and $f_i^{t \to v}$ to the fusion as the probabilities of them belonging to the image feature distribution:

$$p_i^v = p(f_i^v, \mu_v, \sigma_v), \quad p_i^{t \to v} = p(f_i^{t \to v}, \mu_v, \sigma_v), \tag{10}$$

and the contributions of $f_i^t$ and $f_i^{v \to t}$ to the fusion as the probabilities of them belonging to the text feature distribution:

$$p_i^t = p(f_i^t, \mu_t, \sigma_t), \quad p_i^{v \to t} = p(f_i^{v \to t}, \mu_t, \sigma_t). \tag{11}$$

A few examples are demonstrated in Fig. 3 for illustration. Then, we fuse the image and text features of each sample by feeding them into a shared three-layer MLP $F_{v+t}(\cdot)$ after concatenation and obtain the overall fused feature $M_i$ by the weighted sum.

$$\begin{aligned}M_i = &\lambda_i^v \lambda_i^t \cdot (p_i^v p_i^t) \cdot F_{v+t}([f_i^v, f_i^t]) \\ &+ \lambda_i^v \cdot (p_i^v p_i^{v \to t}) \cdot F_{v+t}([f_i^v, f_i^{v \to t}]) \\ &+ \lambda_i^t \cdot (p_i^{t \to v} p_i^t) \cdot F_{v+t}([f_i^{t \to v}, f_i^t]).\end{aligned} \tag{12}$$

Through this process, we explicitly reduce the contributions of low-quality modalities to the fusion, enabling reliable fusion for potential low-quality modalities.

During training, the parameters of encoders are gradually changing, resulting in smooth shifting of the feature distributions. To

**Table 1: Model performances under modality-fixed disruption. We report the ACC/F1 scores of models under C, D, and C+D settings on MVSA-S, MVSA-M, and TumEmo. The highest result is highlighted in bold.**

| Disrupted Modality | Method | MVSA-S | | | MVSA-M | | | TumEmo | | |
|---|---|---|---|---|---|---|---|---|---|---|
| | | C | D | C+D | C | D | C+D | C | D | C+D |
| Image | HSAN [37] | 70.5/69.7 | 69.8/69.6 | 70.0/69.5 | 67.5/65.6 | 66.2/64.1 | 66.6/64.3 | 63.5/63.3 | 62.5/62.4 | 62.9/62.8 |
| | MVAN [41] | 67.7/67.4 | 66.5/66.0 | 66.3/66.2 | 66.9/64.8 | 66.0/63.7 | 66.4/64.2 | 60.7/60.6 | 60.1/60.0 | 60.4/60.4 |
| | MGNNS [43] | 71.9/71.8 | 71.6/70.9 | 71.6/71.3 | 69.4/66.3 | 68.6/65.7 | 69.1/66.2 | 65.2/65.1 | 63.8/63.6 | 64.1/64.0 |
| | CLMLF [17] | 69.4/69.0 | 67.7/67.8 | 68.4/68.1 | 67.0/65.3 | 66.4/64.3 | 66.7/65.0 | 62.4/62.3 | 61.8/61.5 | 62.2/62.1 |
| | MVCN [35] | 70.3/69.9 | 69.3/69.2 | 69.9/69.4 | 68.1/66.0 | 67.3/64.9 | 67.6/65.3 | 63.7/63.6 | 62.9/62.9 | 63.3/63.3 |
| | **DRF (Ours)** | **74.5/74.4** | **73.4/73.1** | **73.8/73.6** | **71.0/68.2** | **70.0/67.5** | **70.3/67.9** | **68.4/68.2** | **67.2/67.2** | **67.9/67.7** |
| Text | HSAN [37] | 64.9/64.3 | 64.1/63.3 | 64.6/64.2 | 64.4/61.6 | 62.9/60.7 | 63.6/61.4 | 48.8/48.5 | 47.5/47.4 | 48.2/48.0 |
| | MVAN [41] | 63.0/62.3 | 62.4/62.2 | 62.8/62.5 | 64.1/60.9 | 62.9/60.0 | 63.5/61.7 | 45.3/45.2 | 44.4/44.0 | 44.8/44.7 |
| | MGNNS [43] | 66.1/65.6 | 64.7/64.5 | 65.5/65.2 | 64.8/62.5 | 63.5/61.8 | 64.1/62.3 | 52.6/52.7 | 50.4/50.4 | 51.5/51.3 |
| | CLMLF [17] | 64.3/63.6 | 63.1/62.8 | 63.7/63.4 | 63.8/61.2 | 62.5/60.4 | 63.3/60.7 | 48.1/48.0 | 46.9/46.7 | 47.0/46.9 |
| | MVCN [35] | 65.3/65.0 | 64.6/64.5 | 65.0/64.7 | 64.4/62.1 | 63.3/61.4 | 63.8/61.9 | 50.5/50.3 | 49.2/49.2 | 49.8/49.7 |
| | **DRF (Ours)** | **69.4/69.4** | **68.1/68.0** | **68.5/68.3** | **67.9/66.5** | **67.2/64.8** | **67.3/66.2** | **61.6/61.4** | **59.2/59.1** | **60.9/61.0** |

**Table 2: Statistics of datasets.**

| Dataset | Total | Train | Val | Test |
|---|---|---|---|---|
| MVSA-S [25] | 4511 | 3608 | 451 | 452 |
| MVSA-M [25] | 17024 | 13618 | 1703 | 1703 |
| TumEmo [41] | 195265 | 156217 | 19524 | 19524 |

track it, we need to progressively update the feature queues with the features from the latest encoders. Meanwhile, we hope to retain the capability of the feature distributions to distinguish modalities of different qualities. To satisfy both requirements, we update the queues with the encoded features of the current sample that exhibit correlations with their respective feature distributions. Specifically, take image $v_i$ as an example, we enqueue $p_i^v$ to $Q_v$ if its probability of belonging to the image feature distribution is larger than the mean of the probabilities of features in $Q_v$:

$$p_i^v > \frac{1}{L} \sum_{j \in q_v} p(f_j^v, \mu_v, \sigma_v). \quad (13)$$

The update strategy for the text feature queue $Q_t$ is similar.

### 3.5 Classification and Optimization

For sentiment prediction, we feed the overall fused feature $M_i$ into a fully connected layer followed by a softmax layer:

$$\hat{Y}_i = softmax(WM_i + b), \quad (14)$$

where $W, b$ are trainable parameters of the fully connected layer, $\hat{Y}_i$ is the predicted probabilities of $S$ sentiment categories. We denote the predicted probability for $k$-th category as $\hat{y}_i^k$, and constrain the classification by a cross-entropy loss:

$$\mathcal{L}_{cls} = -\sum_{k=1}^{S} y_i log(\hat{y}_i^k). \quad (15)$$

To this end, the joint optimization objective for all parameters is:

$$\mathcal{L} = \mathcal{L}_{dis} + \mathcal{L}_{rec} + \mathcal{L}_{cls}. \quad (16)$$

## 4 EXPERIMENT

### 4.1 Dataset Preparations

We carry out experiments on three publicly available MSA datasets. The statistics of them are presented in Table 2. **MVSA-S** and **MVSA-M** [25] are two Twitter datasets annotated by sentiment polarities: {*positive, neutral, negative*}. We pre-process their samples following Xu and Mao [38]. **TumEmo** [41] is a Tumblr dataset annotated according to the emotions of tags. It has 7 emotion categories: {*angry, bored, calm, fear, happy, love, sad*}. We follow the pre-processing of Yang *et al.* [41] for a fair comparison. We report the accuracy score (**ACC**) and F1 score (**F1**) for all three datasets.

To evaluate the robustness of models to low-quality and missing modalities, we simulate these cases by performing two kinds of disruptions on samples. To simulate low-quality modalities, we corrupt images by randomly masking 40-80% of pixels, and texts by replacing 40-80% of words with [MASK] tokens. To simulate missing modalities, we discard modalities from samples. By referring to related fields [27, 34, 54], we incorporate two disruption strategies for a systematical evaluation: modality-fixed disruption and modality-random disruption. In **modality-fixed disruption**,

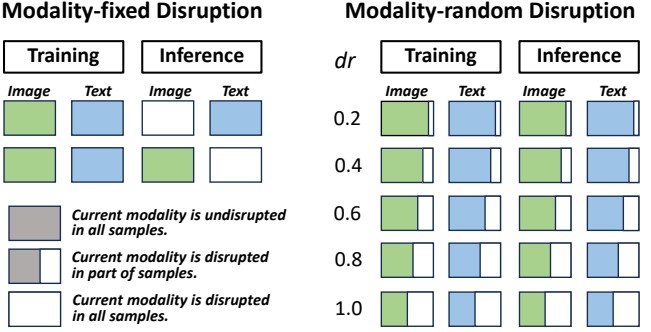

**Figure 4: Illustration of modality-fixed disruption and modality-random disruption strategies.**

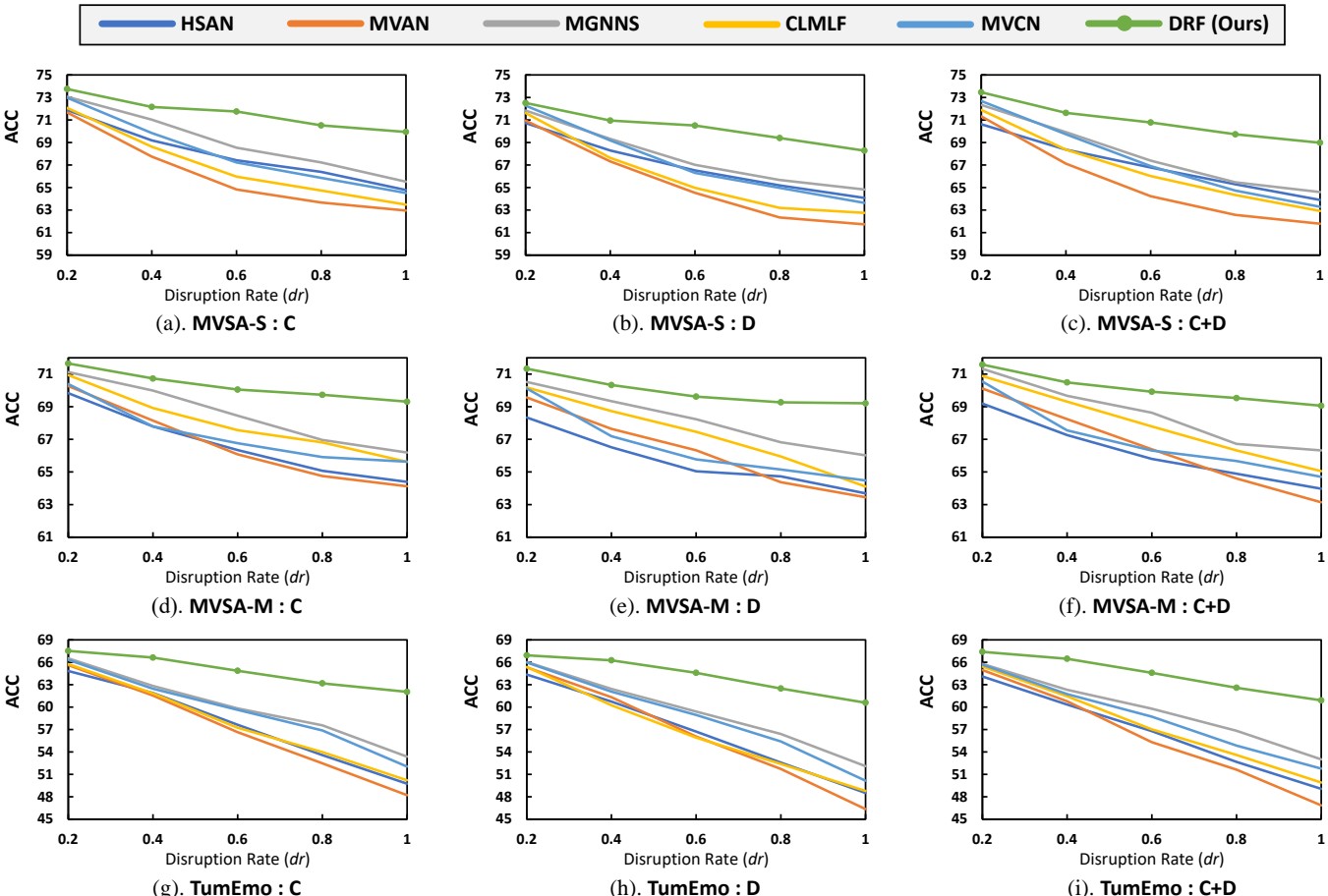

Figure 5: Model performances under modality-random disruption. We report ACC scores of models under C, D, and C+D settings on MVSA-S, MVSA-M, and TumEmo.

we do not interfere with the training process and disrupt a fixed modality for all samples during inference. In **modality-random disruption**, we disrupt a random modality for a pre-defined ratio of samples in both training and inference. At least one modality in each sample is guaranteed to be undisrupted, and reliable for the sentiment prediction. We use the disruption ratio ($dr$) to represent the ratio of samples disrupted and conduct experiments for $dr \in \{0.2, 0.4, 0.6, 0.8, 1.0\}$. We illustrate the two strategies in Fig. 4. For each strategy, we investigate three settings: only corrupts modalities (**C**), corresponding to only introducing low-quality modalities; only discards modalities (**D**), corresponding to only introducing missing modalities; and corrupts and discards modalities half-to-half (**C+D**), corresponding to introducing both low-quality and missing modalities.

### 4.2 Implementation Details

For the image encoder, we adopt Vision Transformer [8] with a patch size of 16, and resize images to 224 × 224. The obtained image features are $d_v = 768$ dimensions. For text, we adopt Bert [7] to obtain text features with the same $d_t = 768$ dimensions. These settings are consistent with the recent SOTA method MVCN

[35] for a fair comparison. We set the mini-batch size to 16 and queue size $L$ to 512. We train the model for 30 epochs with AdamW optimizer. The initial learning rate is set to 2e-5 for image and text encoders and 2e-4 for the rest of the parameters. The learning rates are decayed to 1e-6 in the cosine schedule.

### 4.3 Compared Methods

We compare DRF with a series of SOTA MSA methods to comprehensively validate its effectiveness in robust sentiment classification of image-text pairs. We present brief introductions for the compared methods below. For methods incapable of receiving input with missing modalities, we pad images with blank pixels and texts with [MASK] tokens.

**HSAN** [37] employs image captions to extract image features and concatenates them with text features for sentiment prediction. We reproduce it by replacing its text encoder with a more advanced BERT [7].

**MVAN** [41] separately encodes the object and scene features in images, and interactively models their dependencies with the text features through a memory network.

**Table 3: Ablation study of components under modality-fixed disruption on MVSA-S and TumEmo. Sample-based recovery and distribution-based recovery are the two kinds of supervision on the modality converters introduced in Section 3.3. Gaussian distribution probability is adopted to estimate the quality of modalities. Image-text expansion is the process of expanding each sample into three. They are from Section 3.4. Distribution constraint encourages the compactness in feature distributions and separation between feature distributions, computed by Eq. (3). Experiments for separate components are conducted independently.**

| Disrupted Modality | Method | MVSA-S | | | TumEmo | | |
|---|---|---|---|---|---|---|---|
| | | C | D | C+D | C | D | C+D |
| **Image** | **DRF** | **74.5/74.4** | **73.4/73.1** | **73.8/73.6** | **68.4/68.2** | **67.2/67.2** | **67.9/67.7** |
| | w/o Sample-based Recovery | 73.7/73.4 | 72.1/72.0 | 72.6/72.1 | 68.1/67.9 | 66.0/65.9 | 67.0/67.0 |
| | w/o Distribution-based Recovery | 73.2/72.5 | 71.5/71.2 | 72.2/71.6 | 67.7/67.6 | 65.5/65.6 | 66.7/66.6 |
| | w/o Gaussian Distribution Probability | 71.9/71.7 | 72.7/72.3 | 72.3/72.1 | 65.0/64.7 | 66.6/66.7 | 65.8/65.8 |
| | w/o Image-text Pair Expansion | 72.4/72.2 | 68.3/67.1 | 71.0/70.6 | 66.6/66.5 | 62.8/62.7 | 64.6/64.4 |
| | w/o Distribution Constraint | 74.0/73.8 | 72.5/72.0 | 73.5/73.2 | 67.9/67.9 | 66.3/66.2 | 67.1/67.1 |
| **Text** | **DRF** | **69.4/69.4** | **68.1/68.0** | **68.5/68.3** | **61.6/61.4** | **59.2/59.1** | **60.9/61.0** |
| | w/o Sample-based Recovery | 68.5/68.3 | 66.7/66.4 | 67.5/66.8 | 60.2/60.0 | 57.8/57.7 | 58.9/58.8 |
| | w/o Distribution-based Recovery | 68.5/68.4 | 65.8/65.2 | 67.0/66.9 | 60.4/60.4 | 57.5/57.6 | 59.0/58.8 |
| | w/o Gaussian Distribution Probability | 67.1/66.7 | 67.5/67.5 | 67.3/67.0 | 58.8/58.7 | 58.4/58.4 | 58.6/58.6 |
| | w/o Image-text Pair Expansion | 67.7/67.2 | 65.0/64.8 | 66.2/65.9 | 59.3/59.2 | 53.1/53.0 | 56.2/56.3 |
| | w/o Distribution Constraint | 68.7/68.5 | 67.2/67.0 | 67.9/67.6 | 61.3/61.2 | 58.2/58.3 | 59.5/59.5 |

**Table 4: Model performances without disruption. We report ACC/F1 scores of models on MVSA-S, MVSA-M, and TumEmo. The highest result is highlighted in bold, and the second-highest result is underlined.**

| Method | MVSA-S | MVSA-M | TumEmo |
|---|---|---|---|
| HSAN [37] | 69.9/66.9 | 68.0/67.8 | 63.1/54.0 |
| MVAN [41] | 73.0/73.0 | 72.4/**72.3** | 66.5/63.4 |
| MGNNS [43] | 73.8/72.7 | **72.5**/69.3 | 66.7/66.7 |
| CLMLF [17] | 75.3/73.5 | 71.1/68.6 | 68.1/68.0 |
| MVCN [35] | 76.1/74.6 | 72.1/70.0 | 68.4/68.4 |
| **DRF (Ours)** | **76.5/75.9** | 72.2/70.4 | **69.6/69.6** |

**MGNNS** [43] first introduces graph neural network into MSA, which captures the global co-occurrence characteristics in texts and images, enabling global-aware modality fusion.

**CLMLF** [17] fuses modalities based on Transformer-Encoder [32] to facilitate token-level alignments between modalities. It also proposes two contrastive learning tasks aiding in learning common sentiment features.

**MVCN** [35] tackles the modality heterogeneity from three views: (1). it proposes a sparse attention mechanism to filter out redundant visual features; (2). it restrains representations to calibrate the feature shift; (3) it alleviates the uncertainty in annotations through an adaptive loss calibration.

## 4.4 Comparision with the State-Of-The-Art

### 4.4.1 *Modality-fixed Disruption*. 
The comparison under the strategy of modality-fixed disruption is displayed in Table 1. DRF

consistently achieves the highest results across all cases. It indicates that compared with current methods, DRF is more robust to both low-quality and missing modalities through explicit modeling of modality qualities and building inter-modal mapping relationships. The advantages of DRF under the disruption of texts are more significant. We conjecture that other methods depend more on texts than images due to the higher information density of texts [30]. Subsequently, the corruption and discarding of texts results in severe degeneration of their performances. In contrast, DRF alleviates those influences by flexibly adjusting the contribution of texts and recovering the absent text features.

### 4.4.2 *Modality-random Disruption*.
The results under different disruption rates of modality-random disruption are demonstrated in Fig. 5. As the disruption rate increases from 0.2 to 1.0, the accuracy of DRF is much more stable than other methods. Under the setting of both corruption and disruption (C+D), the accuracy of previous MSA methods drops 6.72%-9.53% on MVSA-S, 5.00%-6.97% on MVSA-M, 12.78%-18.11% on TumEmo, indicating that the modules they devise based on prior knowledge are less effective under disruptions. For instance, MGNNS might be misled by the frequent occurrences of [MASK] tokens and bland pixels, and MVCN might suffer from inaccurate sentimental centroids caused by the disrupted modalities. Under the same setting, the accuracy of DRF only drops 4.48% on MVSA-S, 2.52% on MVSA-M, and 6.50% on TumEmo. These results suggest that the sample and distribution-based recovery and quality-aware fusion facilitate the robustness of DRF to low-quality and missing modalities during both training and inference phases.

### 4.4.3 *Without Disruption*.
The comparison in the regular MSA task without disruption is reported in Table 4. DRF still achieves competitive performances against other methods. We attribute this

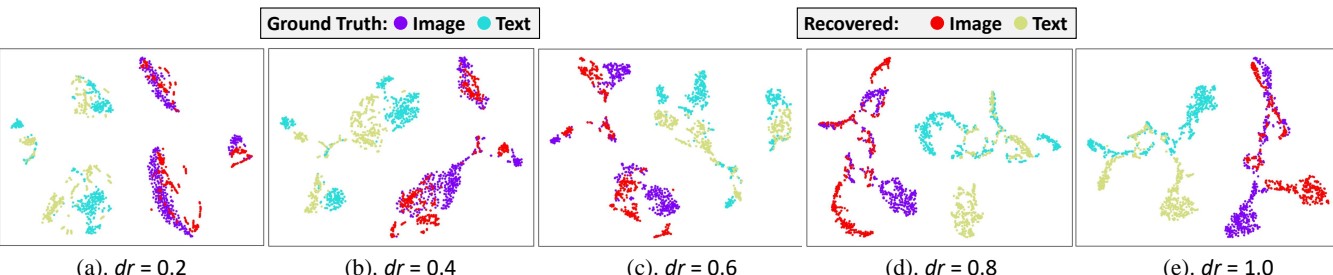

(a). *dr* = 0.2  (b). *dr* = 0.4  (c). *dr* = 0.6  (d). *dr* = 0.8  (e). *dr* = 1.0

**Figure 6: Visualization of image and text features on the MVSA-S test set under different disruption rates of modality-random disruption. Features are projected to 2D space by t-SNE [31].**

to two reasons. Firstly, image-text pairs naturally contain modalities of different qualities. Explicitly quantifying those qualities is beneficial for the reliable fusion of modalities. Secondly, DRF learns the mapping relationships between modalities based on samples and distributions, which promotes more comprehensive information interactions between modalities.

## 4.5 Abalition Study

To validate the effectiveness of each key component in our method, we conduct ablation experiments under modality-fixed disruption in Table 3. From the results, we can derive the following conclusions. Firstly, both the sample-based recovery and distribution-based recovery bring performance improvements to the model, indicating that they are conducive for modality converters to learn local and global mapping relationships between modalities. Secondly, the Gaussian distribution probability and image-text pair expansion significantly facilitate the robustness of the model to low-quality modalities. It emphasizes the effectiveness of explicitly estimating modality qualities and feature fusion based on qualities. Thirdly, the image-text pair expansion also promotes the capability of the model to recover missing modalities under modality-fixed disruption. We conjecture that it introduces the sentiment prediction for recovered samples into the training process, which benefits the similar process during inference. Fourthly, the distribution constraint results in performance gains under both low-quality and missing modalities, verifying the benefits of tightening each distribution and separating different distributions. Finally, combining those components leads to the best performance, proving that they complement each other.

## 4.6 Qualitative Analysis

*4.6.1* **Feature Recovery Visualization.** To intuitively present the efficacy of two recovery tasks in Section 3.3, we visualize the image and text features recovered by DRF under modality-random disruption with disruption rate increases from 0.2 to 1.0. We project the samples of the MVSA-S test set into 2D space by t-SNE [31] and display them in Fig. 6. Under low disruption rates, the recovered features closely adhere to the ground truth features. It demonstrates that DRF learns accurate mapping relationships between modalities based on the local guidance of sample-based recovery and global guidance of distribution-based recovery. As the disruption rate increases, the sample-based recovery gradually becomes unavailable, yet DRF can still recover features with distributions similar to the

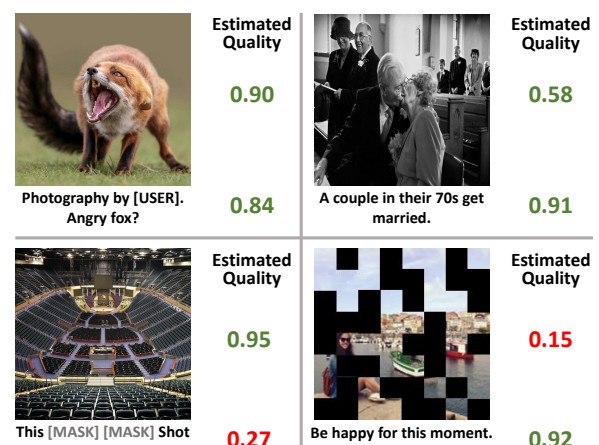

**Figure 7: The estimated modality qualities of undisrupted samples (above) and corrupted samples (below).**

ground truth features. It proves the effectiveness of distribution-based recovery and emphasizes its necessity under high disruption rates.

*4.6.2* **Modality Quality Visualization.** In Fig. 7, we display the estimated modality qualities of four samples, including two undisrupted and two corrupted. For the undisrupted samples, DRF assigns relatively high qualities for all four modalities, with the black-white image receiving the lowest estimated quality. For the corrupted samples, DRF correctly assigns relatively low qualities for the corrupted modalities. These results validate the effectiveness of estimating modality quality based on the distributions.

## 5 CONCLUSION

In this paper, we focus on robust multimodal sentiment analysis of image-text pairs with possible low-quality and missing modalities. These issues are prevalent in real-life applications yet under-explored by previous studies in this subfield. We propose a method called DRF to handle these issues in a unified framework. It approximates the feature distributions by feature queues and leverages them to simultaneously provide global guidance for feature recovery as well as quality estimation of each modality for feature fusion. Through comprehensive experiments, we demonstrate the effectiveness and robustness of the proposed DRF.

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
