# OpenReview forum: "Robust Multimodal Sentiment Analysis of Image-Text Pairs by Distribution-Based Feature Recovery and Fusion"
_acmmm.org/ACMMM/2024/Conference — MM2024 Poster_

### Official Review · Reviewer_LBkC · 2024-05-20

**Rating:** 3
**Confidence:** 3

**Summary:**

This paper mainly proposes a distribution-based feature recovery and fusion (DRF) method for robust multi-modal sentiment analysis of image-text pairs. A feature queue is maintained for each modality to approximate its feature distribution, while low-quality and missing modalities are simultaneously handled in a unified framework. In the experiments, two destruction strategies of destroying and discarding certain modes in the samples are adopted to simulate low-quality and missing modes in various real-world scenarios. Experiments on three publicly available image-text pair datasets demonstrate the general improvement of DRF over SOTA methods under both strategies.

**Strengths:**

This paper focuses on robust MSA for image-text pairs with low quality and missing modalities, which are common problems in real-world scenarios. This article is the first attempt to explore the robustness of models in this subfield. The amount of experiments in this paper is very sufficient, and it also achieves good performance.

**Limitations:**

1.The symbols in some formulas are not noted. It is recommended to explain them clearly under the formulas.
2.The two main challenges mentioned in the introduction, the first problem is the alignment burden of the unaligned model, and the second problem is that the user-generated nature of social media posts leads to frequent mismatches between images and text, which are also inherently aligned question, it is recommended to rephrase these two main challenges.

**Suitability:**

2

---

### Official Review · Reviewer_bb26 · 2024-05-22

**Rating:** 3
**Confidence:** 4

**Summary:**

This paper proposes a Distribution-based feature Recovery and Fusion (DRF) method for robust multimodal sentiment analysis. DRF maintains a feature queue for each modality to approximate their distributions, enabling the handling of low-quality and missing modalities within a unified framework. By estimating modality quality, DRF reduces the influence of low-quality data and recovers missing modalities through inter-modal mapping. Experiments using disruption strategies on three image-text datasets demonstrate DRF's effectiveness, showing universal improvements over state-of-the-art methods.

**Strengths:**

1. The manuscript is well-structured, the figures are well-designed and effectively illustrate key concepts, enhancing the reader's understanding of the proposed approach and its results.

2. The proposed method achieves state-of-the-art (SOTA) performance on the selected datasets, demonstrating its efficacy and robustness. The experiments are relatively comprehensive, validating the method's generalizability and effectiveness across different conditions.

3. The algorithms are efficient and primarily based on Gaussian distribution, which is advantageous in terms of saving computational resources.

**Limitations:**

1. The manuscript contains numerous long sentences, which can make it challenging for readers to follow the main ideas. Breaking down these lengthy sentences into more concise and focused statements would improve readability and enhance the clarity of the presentation.

2. The algorithm for modality recovery relies on forcing modalities to be close based on Euclidean distance, which lacks interpretability regarding the feature recovery process. This approach may not adequately capture the nuanced relationships between different modalities, and a more interpretable method could provide deeper insights into how features are recovered and fused.

3. This study does not adequately address the semantic gap between different modalities, which is a crucial aspect of multimodal learning. Incorporating methods to bridge the semantic differences between modalities would enhance the robustness and effectiveness of the proposed approach.

4. Although the manuscript proposes a series of algorithms for modality recovery and quality estimation, these algorithms appear to have limited relevance to the extraction of sentiment clues within different modalities for this task. The focus on recovery and quality estimation may overshadow the critical aspect of identifying and integrating sentiment-related features, which is essential for effective multimodal sentiment analysis.

5. The manuscript does not explain the rationale behind the selection of the feature queue size 𝐿.  Specifically, it is unclear why the queue size is set to 512. A detailed justification for this choice is necessary to understand its impact on the performance and computational efficiency of the proposed method.

**Suitability:**

3

---

### Official Review · Reviewer_RkT8 · 2024-05-23

**Rating:** 5
**Confidence:** 3

**Summary:**

The authors propose a distribution-based feature recovery and fusion method for robust sentiment analysis tasks of image-text pairs. In real-world applications, the images and texts of posts may be corrupted or missing, leading to frequent occurrences of low-quality and missing modalities. Focusing on these two issues, this paper summarize the shortcomings of the existing methods, and the motivation behind the proposed method is explained. At last, experimental results prove the effectiveness of proposed method in robust multimodal sentiment analysis.

**Strengths:**

1.The author identifies the limitations of current multimodal sentiment analysis methods from a real-world perspective and proposes their own ideas for improvement. This has practical value and significance.
2.The approach presented in this paper is sufficiently innovative. The authors build mapping relationships between image and text through two modality converters, which can address the issues of low quality and missing modalities simultaneously. By introducing feature distribution recovery, the mismatch between images and text in social media posts is effectively alleviated. This make sure the converter learns correctly.
3.The author demonstrates strong writing ability and presents a well-structured paper. The experimental design is thorough, evaluating the robustness of each model as well as their performance on standard datasets. This study confirms that the proposed method not only exhibits good robustness, but also delivers strong results on standard data.

**Limitations:**

1.The experimental design was adequate, however, there was a lack of novel methods for comparison. Incorporating newer methods for comparison would better demonstrate the advanced nature of the method.
2.The authors adopt the Gaussian distribution to estimate the quality of each modality. Generally, low-quality features are distributed on sides of the Gaussian distribution, but the features on sides are not necessarily low-quality. This results in modalities that are rich in information but slightly different from the average distribution being ignored.
3.Figure 6 is the visualized results of image and text features. The authors show that as the disruption rate increases, the DRF can recover features closer to the ground truth. But that's not obvious from the figure. Authors should provide sample-based recovery results for comparison.
4.Figure 7 does not show the advantages of modal fusion proposed in this paper. When modeling a low-quality modal data, it is crucial to not only demonstrate its modal quality but also the recovered modal quality from the other modality. In this way, it is helpful to display the improvement of modal fusion through modal quality estimation.
5. Please provide the code for reproduction.

**Suitability:**

3

---

### Official Review · Reviewer_hgTg · 2024-05-24

**Rating:** 3
**Confidence:** 3

**Summary:**

This paper focus on robust Image-text sentiment analysis of image-text pairs for the low-quality and missing modalities, and propose a novel Distribution-based feature Recovery and Fusion method for the challenges.

**Strengths:**

This paper first introduces the low-quality and missing modalities challenges for robust Image text sentiment analysis area.
The paper is well-organized, and the performance is good.

**Limitations:**

1. Missing literature review in Robust Multimodal sentiment analysis (there exist many robust MSA methods before). Although few of existing for exactly Image text sentiment analysis, the reviewer still suggest the author to include introduction and even performance comparison with previous works in Robust MSA field, such as work [1,2,3,4,5].

[1] Transformer-based feature reconstruction network for robust multimodal sentiment analysis
[2] Efficient multimodal transformer with dual-level feature restoration for robust multimodal sentiment analysis
[3] MM-Align: Learning Optimal Transport-based Alignment Dynamics for Fast and Accurate Inference on Missing Modality Sequences]
[4] Missing modality imagination network for emotion recognition with uncertain missing modalities
[5] Gcnet: Graph completion network for incomplete multimodal learning in conversation
[6] Noise imitation based adversarial training for robust multimodal sentiment analysis

2. One of the Limitations is all baselines (except for MVCN) are proposed before 2022, the reviewer suggest the author to add more recent baselines for comparison. And how is the performance of directly using unimodal models for low-quality and missing modalities instances?

3. Moreover, as there exist many MLLMs (Multimodal Large Language Models) and achieving promising results in most multimodal scenarios, additional comparison with current MLLMs will further make the contribution more sufficient.

Other Questions.

1. CLMLF utilizes a contrastive learning method with positive and negative labels to conduct experiments. However, TumEmo is a multi-class dataset. The author should describe the details of how the CLMLF is conducted on the TumEmo dataset.
2. The MVSA dataset is highly imbalanced. I am curious about how each experiment is conducted. Is it run multiple times to obtain an average, or is it performed using k-fold cross-validation?

**Suitability:**

3

---

### Meta-Review · Area_Chair_YLQz · 2024-07-01

**Recommendation:** Accept (Poster)
**Confidence:** 4

**Metareview:**

Most of the reviewers lean towards acceptance. The authors have addressed most of the reviewers' concerns. However the average score is BA and there still are concerns.